# BRIDGING BETWEEN POOL- AND STREAM-BASED ACTIVE LEARNING WITH TEMPORAL DATA COHERENCE

## ABSTRACT

Active learning (AL) reduces the amount of labeled data needed to train a machine learning model by choosing intelligently which instances to label. Classic pool-based AL needs all data to be present in a datacenter, which can be challenging with the increasing amounts of data needed in deep learning. However, AL on mobile devices and robots like autonomous cars can filter the data from perception sensor streams before it even reaches the datacenter. In our work, we investigate AL for such image streams and propose a new concept exploiting their temporal properties. We define three methods using a pseudo uncertainty based on loss learning (Yoo & Kweon, 2019). The first considers the temporal change of uncertainty and requires 5% less labeled data than the vanilla approach. It is extended by the change in latent space in the second method. The third method, temporal distance loss stream (TDLS), combines both with submodular optimization. In our evaluation on an extension of the public Audi Autonomous Driving Dataset (Geyer et al., 2020) we outperform state-of-the-art approaches by using 1% fewer labels. Additionally, we compare our stream-based approaches with existing approaches for AL in a pool-based scenario. Our experiments show that, although pool-based AL has more data access, our stream-based AL approaches need 0.5% fewer labels.

## 1 INTRODUCTION

Active learning (AL) is a technique to minimize the labeling effort, in which a machine learning model chooses the data to be labeled by itself. It can be divided into two main scenarios, pool-based and stream-based AL (Settles, 2010). Pool-based AL is a cyclic process of selecting batches of the most promising samples from a pool of data based on a query function. The model is retrained after the selection to start the next iteration of the AL cycle. The data pool is stored such that all samples are always accessible. In contrast, stream-based AL assumes an inflow of samples as a stream and the model decides if a sample should be saved and labeled or disposed. In classic stream-based AL the model is trained with each selected sample (Settles, 2010). However, in deep learning samples are usually selected in batches, due to the long training time of the models. This comes with the risk of selecting samples with an equal information gain. Most approaches ignore this fact or solve it by using a small selection batch size.

Besides the scenarios, the selection method, also called querying strategy, is another important factor of AL methods. There are three main categories of AL algorithms: uncertainty-based, diversity-based and learning-based AL (Ren et al., 2022). The first group are uncertainty-based AL methods, including for example Monte Carlo (MC) dropout methods (Gal & Ghahramani, 2016) or methods approximating the uncertainty by using ensembles (Beluch et al., 2018). The second group are diversity-based methods like Coreset (Sener & Savarese, 2018) or diverse embedded gradients (Ash et al., 2020). These methods select samples based on the dataset coverage. The third group are learning-based approaches. These methods, like loss learning (Yoo & Kweon, 2019), train an additional model, which either predicts a value, determining the usefulness of a sample, or decides if a sample should be selected directly. Recent approaches from this category often include unlabeled data for unsupervised training. Other approaches taking diversity into account usually perform an optimization, which requires constant access to the complete labeled and unlabeled dataset. This decreases the number of needed samples as intended, but the access to unlabeled data makes the transfer to a stream-based scenario impossible.

A large body of research in the perception domain focuses on pool-based AL, which requires the transfer of all data to a datacenter. Especially in autonomous driving AL is already an important research topic (Feng et al., 2019) (Hekimoglu et al., 2022). However, the data logistics and data preparation limits the possibilities to apply and scale this approach to open world perception problems, where a lot of data is required. These perceptions task including autonomous driving and robotic perception and environmental sensing. In contrast to pool-based AL, stream-based AL can run directly on mobile devices used in these applications and enables data collection through a large number of agents without a prior transfer to the data center. By performing AL on a mobile robot, it can be applied on temporally coherent camera streams directly, which reduces preprocessing efforts. Based on these considerations we focus on stream-based AL for temporally coherent data.

Our contribution can be summarized as follows: We suggest a novel concept of incorporating temporal information into AL, especially stream-based AL. Our concept exploits the temporal change of uncertainty and distance in latent space. Based on this we propose three methods and compare them with state-of-the-art methods in a classification task; the most commonly used task to benchmark AL. We evaluate our methods against other state-of-the-art methods. Therefore, we create a operational domain detection dataset by adding scene annotations to the Audi Autonomous Driving Dataset (A2D2) (Geyer et al., 2020). Further, we give an overview of the necessary steps to transform a pool-based scenario in a stream-based scenario and perform, to the best of our knowledge, the first direct comparison between stream-based and pool-based AL methods.

## 2 RELATED WORK

While a lot of authors did great research in the field of pool-based AL, stream-based AL has become unpopular with the rise of deep learning. However, the number of vision sensors receiving constant data streams is increasing, so will the cost of transferring these data to a datacenter. This makes research of stream-based AL techniques interesting, as not all data can be transferred to the datacenter to perform pool-based AL.

### 2.1 POOL-BASED ACTIVE LEARNING

Sener & Savarese (2018) defined AL as a core set selection problem. The authors aim to select samples that minimize the maximum distance to other not selected points. In this way, it can be formulated as a K-center problem. Solving this is quite costly, so the authors suggested to use a greedy algorithm to approximate the K-center problem. The method will be further denoted as Coreset. In Bayesian active learning with diverse gradient embedding (Badge) Ash et al. (2020) the diversity idea has been extended by taking the prediction uncertainty into account. The authors combined a distance representation of the latent space with pseudo labels based on the highest one-hot encoded value to generate gradients. These are created for every class such that the dimension of the embedding is higher than in the Coreset (Sener & Savarese, 2018) approach. The optimal set is estimated using greedy optimization algorithms.

An uncertainty-based approach is MC dropout as a Bayesian approximation (Gal & Ghahramani, 2016). The method uses several dropout layers which are active during the prediction phase. By performing multiple forward passes a distribution over the class predictions is generated where the authors applied the mutual information function in order to calculate the uncertainty of the samples. This is often combined with the Bayesian active learning by disagreement (Houlsby et al., 2011) metrics, considering the mutual information of the multiple forward passes. Their approach has been modified by Kirsch et al. (2019) to take the diversity of the selected batch into account by calculating the joint mutual information. With their BatchBald approach, the authors reduced the selected samples with redundant information in a batch. In contrast to sampling-based approaches, loss learning (Yoo & Kweon, 2019) is a learning-based approach that needs only one forward pass. By adding a loss module to specific layers of the prediction network, the authors predicted the network's loss and used it as pseudo uncertainty for sample selection. However, the loss module can only predict a relative loss. The authors showed the flexibility of the approach for several tasks, which makes it quite popular. Novel learning-based methods like variational adversarial active learning (VAAL) (Sinha et al., 2019) use the unlabeled data as well. An autoencoder is trained to learn a latent space representation of the data based on the labeled and unlabeled set. Based on the latent space encoding, a discriminator model is trained to discriminate between labeled and unlabeled data.

The selection is based on the lowest prediction confidence of the discriminator out of the unlabeled dataset predictions. The authors outperformed algorithms like Coreset (Sener & Savarese, 2018) and MC dropout (Gal & Ghahramani, 2016) methods. Kim et al. (2021) extended VAAL (Sinha et al., 2019) by adding a loss prediction module to the task model and the predicted loss to the latent space such that it is included in the input vector of the discriminator model. A more diversity-oriented learning-based approach using unlabeled data is sequential graph convolutional network for active learning (Caramalau et al., 2021). The authors used the distance between the features of the task model to calculate an adjacency matrix for a graph containing labeled and unlabeled data. Based on this matrix a graph neural network is trained. By using message passing the network should predict the nodes' value for being labeled. With this approach, further denoted as CoreGCN, the authors achieved a good performance on classification datasets.

## 2.2 STREAM-BASED ACTIVE LEARNING

Stream-based AL has rarely been used for perception tasks so far, especially with deep learning models. In the field of perception Narr et al. (2016) selected data by using mondrian forests stream-based AL and trained them in an online learning fashion. For non-deep neural network models online and incremental learning (Chiotellis et al., 2018) is often combined with AL as classical stream-based AL models are retrained after each selection. Another challenge is dealing with conceptual drifts (Łukasz Korycki et al., 2019), (Pham et al., 2022) in which the underlying distribution is changing over time. Especially in stream-based AL, the selection is seen as a submodular optimization problem where the value of an added labeled sample is dependent on the labels already present. As solving these problems is computationally expensive, stream-based greedy algorithms are an important field of research (Fujii & Kashima, 2016). A method for solving submodular optimizations problems is Sieve-Streaming++ (Kazemi et al., 2019). The concept of submodular optimization opens many possibilities for future work. Sieve-Streaming++ has been used by Senzaki & Hamelain (2021) explicitly for AL on edge devices (ALED). The authors tried different semi-positive definite kernels with the prediction confidence as a value function. Temporal and stream properties are neglected in their work.

In our work we will neglect online learning and concept drifts and focus on connecting pool- and stream-based AL for perception.

## 2.3 ACTIVE LEARNING ON TEMPORAL DATA

Although temporal coherence is an important property of a stream, this property is only used for pool-based AL in previous works. Bengar et al. (2019) used the object detection false positive (FP), true positive (TP), false negative (FN) and true negative (TN) metrics to build a temporal graph and select samples with energy minimization. As this approach requires ground truth, it can be only used as a theoretical baseline. Besides, the authors provided the SYNTHIA-AL dataset, based on the SYNTHIA (Ros et al., 2016) dataset created for AL purposes. Due to the short snippets and high sampling rate, the dataset mostly targets semantic segmentation or object detection applications. Schmidt et al. (2020) used the object detection classification uncertainty estimated by the entropy over a time horizon. To do so the authors used preceding and succeeding images of each Kitti (Geiger et al., 2012) sample. By using this approach a comparable uncertainty can be estimated, avoiding the usage of ensembles (Beluch et al., 2018) or MC dropout (Gal & Ghahramani, 2016) methods. Nevertheless, the authors only described this approach for pool-based AL. Huang et al. (2018) used temporal information to avoid multiple MC dropout forward passes (Gal & Ghahramani, 2016) for semantic segmentation by combining on forward pass uncertainty prediction with a flow network to calculate the uncertainty as moving average over a time horizon.

Although many topics have been covered, in particular in pool-based AL, temporal properties have only been used to save computation for MC dropout (Gal & Ghahramani, 2016) passes. Temporal properties for stream-based AL still appear to be a relatively unexplored research topic. We want to investigate the change of uncertainty and diversity over time, especially for the seldomly covered stream-based AL.

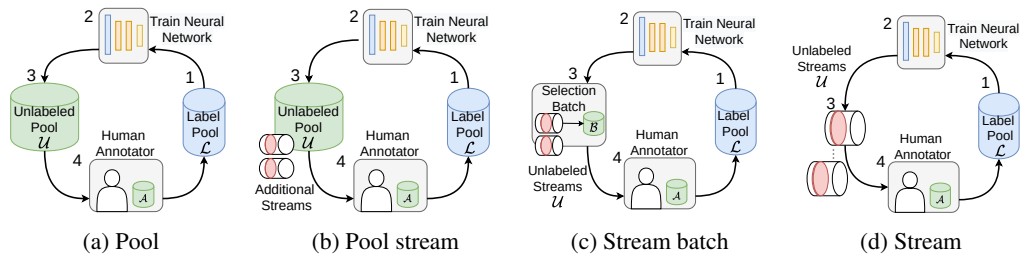

Figure 1: Active learning scenarios from pool to stream

# 3 FROM POOL-BASED TO STREAM BASED-ACTIVE LEARNING

As pool-based and stream-based AL are quite detached, we want to bridge the gap and enable comparisons by adding two intermediate scenarios. Namely the pool stream and stream batch scenario. A collection of the different scenarios is depicted in Figure 1. All scenarios start with a small labeled dataset (1) which is used to train a model (2). In the classic pool-based scenario, shown in Figure 1a, the samples are selected from a constant unlabeled pool (3) and sent to an oracle (4) for labeling. All samples including the unlabeled ones can be seen and used multiple times. The second scenario depicted in Figure 1b, reflects a continuous data selection, in which the unlabeled pool (3) is extended at every cycle. To reflect limited recording and transferring capabilities we change the scenario to a stream-based one. We remove the pool to create the stream batch scenario depicted in Figure 1c, where the current stream is visible for the model only once. In contrast to the classical stream scenario, a batch $\mathcal{B}$ (3) of a maximum size of $b$ can be selected from each stream. In the classic stream scenario depicted in Figure 1d, the samples need to be chosen or disposed immediately. This adds the challenge of defining a threshold function classifying useful and useless samples.

Having introduced the four scenarios, we define four categories for the AL querying strategies and evaluate the possibility of using them in stream scenarios. The first category contains methods evaluating samples individually like loss learning (Yoo & Kweon, 2019), ensembles (Beluch et al., 2018) and MC dropout (Gal & Ghahramani, 2016) methods. These can be used for stream and pool scenarios without any adaptation. The second category describes methods performing a optimization which requires access to all unlabeled data during the optimization. This category contains mostly the diversity-based methods Coreset (Sener & Savarese, 2018), Badge (Ash et al., 2020) and BatchBald (Kirsch et al., 2019) , which cannot be used for stream-based AL directly. However, they can be used if the greedy optimization can be transformed to work on streams. The third category contains methods that use unlabeled data for training, such as CoreGCN (Caramalau et al., 2021) or VAAL (Sinha et al., 2019), and cannot be transferred to stream-based scenarios. The fourth category contains methods that are stream-based, such as the video summarization (Kazemi et al., 2019) and ALED (Senzaki & Hamelain, 2021). As current AL research in perception is focusing on the third category, the number of methods that can be transferred to a stream scenario is quite limited. Only the first and fourth category can be used in all AL scenarios.

# 4 TEMPORAL INFORMATION IN PERCEPTION DATA

Most perception datasets do not contain temporal data. Since these datasets are meant for classification, object detection or semantic segmentation tasks this information is naturally of lower importance for the task at hand. The commonly used datasets Kitti (Geiger et al., 2012) or Cityscapes (Cordts et al., 2016) aim to have a good diversity to be highly generalizable. Classification datasets like Cifar10 (Krizhevsky et al., 2009), which is often used to benchmark AL, are not temporally ordered data streams. Benchmarking AL on these datasets is sub-optimal and only shows potential label savings on datasets that have been manually designed for diversity. Instead, we propose that AL shall be benchmarked on camera or sensor streams directly such that no additional manual work besides labeling is needed.

We create our benchmark dataset based on A2D2 [1] (Geyer et al., 2020) which provides temporally coherent frames structured in different drives. We assign the classification labels urban, highway, country road and construction site describing the driving environment to create an operation domain detection task. This task is important in mobile robotics to estimate if action can be executed safely. The dataset contains several recorded drives in southern Germany, with around 680 frames on average per recording. The data is temporally clustered in the latent space by the nature of the drives, which can be seen in Figure 2. Further details can be found in Appendix A.

## 5 VALUE OF TEMPORAL INFORMATION FOR ACTIVE LEARNING

By defining the drives of the dataset as consecutive order streams, properties like the predictive uncertainty $\sigma_p$ can be represented as function of time $t$. As sampling-based approaches are problematic for stream-based applications, due to increased computational cost, we use a loss module $f_\mathcal{L}$ (Yoo & Kweon, 2019) to estimate the predictive uncertainty. The predicted loss $\hat{\mathcal{L}}$ (pseudo uncertainty) of a sample $x$ can be defined as in Equation 1.

$$\sigma_p \approx \hat{\mathcal{L}}_x = f_\mathcal{L}[x] = f_\mathcal{L}^*[t_x] \qquad (1)$$

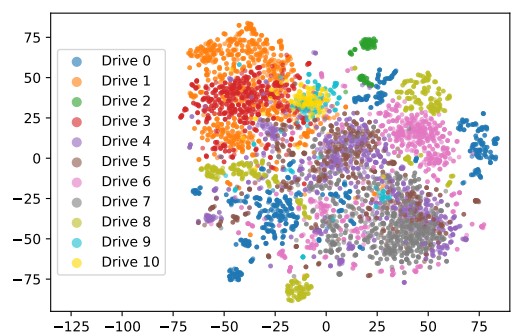

The loss module, as well as latent space representations depend on the current selected training set and are updated after each AL cycle. As time is strictly increasing, the time derivative of the predicted loss $\frac{d}{dt}\hat{\mathcal{L}}$ exists. By taking the

Figure 2: T-SNE analysis with perplexity 30 of the different recorded drives from the training set.

temporal coherence properties of a sensor stream into account and assuming real world (or their simulations) situations, the change between two samples is naturally limited. This effect can be observed in Figure 2, where the different drives form natural clusters. We use the existence of a time derivative to propose our first method[1], Temporal Predicted Loss (TPL):

$$\lambda_i = \left\lVert \frac{d}{dt}\hat{\mathcal{L}} \right\rVert \qquad (2)$$

Based on the selection value $\lambda_i$, we select the sample $i$ with the highest absolute temporal change in the predicted loss, instead of selecting samples with the highest predicted loss for the next AL cycle. By taking the temporal change into account the method can easily filter similar samples as they have a close temporal relation as well as similar uncertainty values. In addition, the method is very sensitive to samples having sudden changes which cause a change in uncertainty, which can be challenging for the model. In Figure 3 we compare the latent space coverage of loss learning and TPL using t-SNE plots. While the vanilla loss learning approach mostly covers only one corner, our approach selects samples all over the latent space.

Our second method, Temporal Distance Loss Change (TDLC), is also taking the diversity of the dataset into account by analyzing the change in the latent space representation of the samples. As Figure 2 shows, the samples of one drive are often grouped in clusters. We want to investigate if the change of distance in latent space is a suitable metric to increase the performance of a selection query. Thus, we formulate Equation 3 to combine the temporal change of the predicted loss $\frac{d}{dt}\hat{\mathcal{L}}_i$ with the temporal change in latent space $\frac{d}{dt}\boldsymbol{f}_i$ scaled by the factor $\delta$ which is set to one. In this equation, $i$ denotes the sample and $\lambda_i$ its selection value. As the magnitude of the learned loss and distance in latent space can be different, we calculate the mean and standard deviation of both values on the fly denoted as $\overline{\text{mean}}$ and $\overline{\text{std}}$ to combine the zero mean unit variance value of both flows.

$$\lambda_i = \frac{\frac{d\hat{\mathcal{L}}_i}{dt} - \overline{\text{mean}}(\frac{d\hat{\mathcal{L}}}{dt})}{\overline{\text{std}}(\frac{d\hat{\mathcal{L}}}{dt})} + \delta \cdot \frac{\frac{d\boldsymbol{f}_i}{dt} - \overline{\text{mean}}(\frac{d\boldsymbol{f}}{dt})}{\overline{\text{std}}(\frac{d\boldsymbol{f}}{dt})} \qquad (3)$$

[1]Will be made available upon acceptance to preserve authors' privacy.

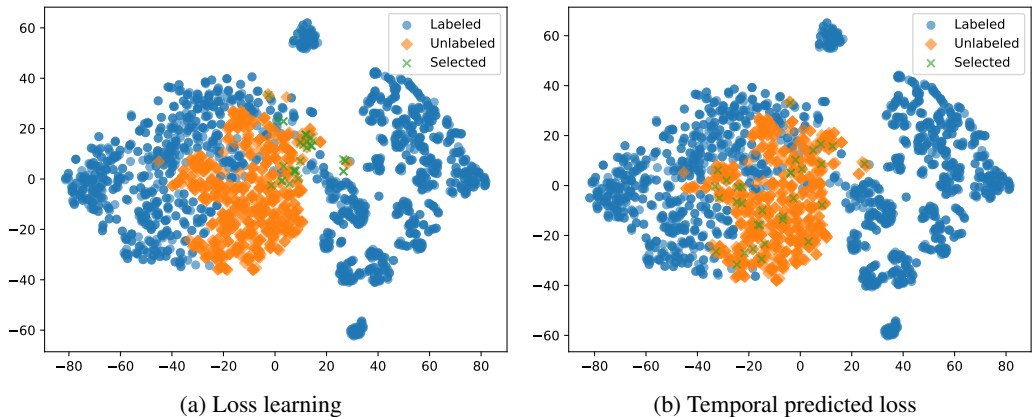

(a) Loss learning

(b) Temporal predicted loss

Figure 3: Latent space comparison between loss learning and temporal predicted loss using t-SNE with a perplexity of 30 for drive 20181108_084007.

The last method is following the idea of submodular optimization, as these methods take the relations inside the batch into account. We base our method on the Sieve-Streaming++ algorithm and follow the idea of (Kazemi et al., 2019) and (Senzaki & Hamelain, 2021) to use the determinant of a positive semi-definite kernel. We choose to evaluate the distance of the selected samples with the dot product of the feature vectors, which have also been evaluated by (Senzaki & Hamelain, 2021). In contrast to the distance matrix used in (Kazemi et al., 2019) the dot product of the vectors takes the direction in latent space into account. The resulting matrix has the squared norm of the selected vectors as the main diagonal, while the other values depend on the orientation of the data points from the origin of the latent space. These mathematical properties lead to a higher diversity, compared to a regular distance matrix. We integrate the temporal change of the predicted loss $\frac{d}{dt}\hat{\mathcal{L}}_i$ of the $i$-th sample with the matrix product of the feature vectors into submodular optimization. The value $\lambda_{\mathbb{J}}$ of the selected set $\mathbb{J}$ with $j$ elements in Equation 4 is to be maximized. Where $j$ is constrained by the batch size $b$ to $j \leq b$. $\boldsymbol{F}^{j \times n}$ denotes the matrix of $j$ latent space feature vectors with $n$ equals the feature dimension. We followed Senzaki & Hamelain (2021) and set the scaling factor $\delta$ to 0.5. This method is further denoted as Temporal Distance Loss Stream (TDLS).

$$\lambda_{\mathbb{J}} = \sum_{i=0}^{j} \left\| \frac{d}{dt}\hat{\mathcal{L}}_i \right\| + \delta \cdot \log(\det(\boldsymbol{F}\boldsymbol{F}^T + \boldsymbol{I}_j)) \tag{4}$$

As ablation study, we replace $\boldsymbol{F}^{j \times n}$ with the gradient embedding based on Ash et al. (2020) $\boldsymbol{E}^{j \times (n \cdot c)}$ with $c$ being the number of classes, further referred as Temporal Embedded Gradient Loss Stream (TEGLS). This adds information on possible loss directions to the optimization.

## 6 EXPERIMENTS AND RESULTS

For our experiments, we use the A2D2 object detection dataset labeled as described in Section 4. We start with an initial training set of two drives with 1674 images. Another nine drives with 4518 images remain unseen to be used as streams in the AL cycles. For the testing and validation set, we use three drives each, with a total of 2776 and respectively 2577 images. All splits and further detail can be found in Appendix A. As the streams differ from each other in length, we use a percentage selection size for each stream instead of a fixed one. At each AL cycle, indicated as marker in the plot, a new stream will be added according to the specified scenario from Figure 1. In the results figures we plot the accuracy over the percentage used from the initial training set and the unlabeled pool. As baselines, we use a neural network trained on the whole training set including all possible selections, as well as a random selection strategy. Besides these two commonly used baselines, we introduce a fixed step selection strategy which is an often used strategy to reduce the number of samples in a recording. We use a ResNet18 (He et al., 2016) model for most experiments as it is the most common model in the related work. Further, we extend the classification head to three

fully connected layers with dropout layers in between, such that it can be compared with sampling methods like BatchBald. For the convolutional layers the pre-trained ImageNet (Deng et al., 2009) weights provided by PyTorch (Paszke et al., 2019) are used. As we noticed a positive effect of the joint loss from the loss learning module, we add this module to all models for a fair comparison. All hyperparameters and model details are listed in Appendix B. After each selection cycle, the model is trained from scratch. As to our knowledge no other comparable datasets are available, we variate the order of streams ingested in the cycles to prove the robustness of our approach. The three tested orders are shown alternately in the figures such that each order is shown twice. The same five seeds are used for the different methods. At first, we compare our methods in the stream-based scenario introduced in Figure 1c and finally we relate them to the pool stream scenario from Figure 1b.

## 6.1 TEMPORAL COHERENCE FOR BATCH DIVERSIFICATION

In our first experiment, we want to show the influence of the temporal relation. In Figure 4 we compare loss learning with TPL in the stream batch scenario from Figure 1c . We show both approaches for the three selection sizes 5%, 10% and 20%. Besides the ResNet18 model, we compare both approaches with the ResNet34 model to prove flexibility. It can be seen that our approach outperforms the vanilla loss learning approach for different models and selection sizes. For all parameters it reaches a higher accuracy score with the same amount of labeled data. For ResNet18 shown in Figure 4a only our approach reaches the performance of the network trained on all data. In Figure 4b our approach clearly reaches the standard deviation region of the fully trained model for a selection size of 20%, while the vanilla loss learning approach reaches it in the last step with 5% additional data selected. The overall performance of the loss learning approach is lower for ResNet34 which influences our approach as well. Qualitatively the increased diversity of our method can be seen in Figure 5b in comparison to loss learning shown in Figure 5a.

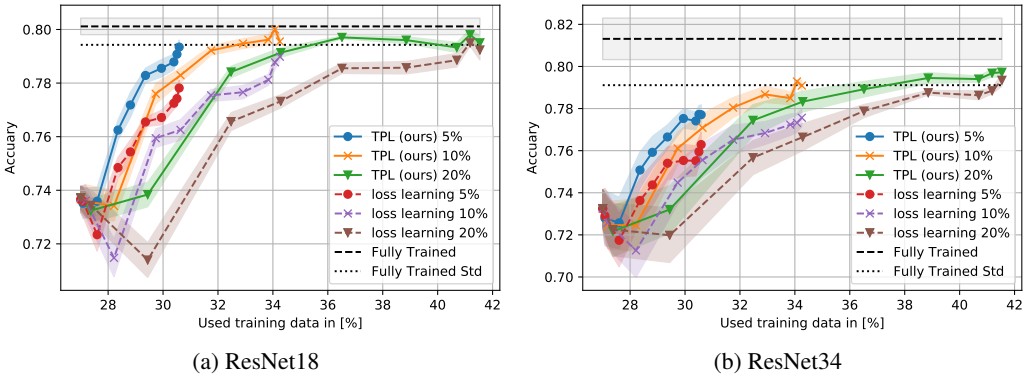

(a) ResNet18          (b) ResNet34

Figure 4: Comparison of loss learning with TPL for different selection sizes and models. Colored areas indicate standard errors. The lower standard deviation of the fully trained model is dotted.

In general, it can be seen that our approach outperforms vanilla loss learning for different selection sizes and models. The effect is reduced with a larger selection size, which was expected. In addition, loss learning seems not the perfect method to estimate the (pseudo) uncertainty of a sample as its performance varies between the models in Figure 4. However other approaches require either multiple models or forward passes, which increases computation cost and is therefore problematic for streams. This is important if the selection is performed on a mobile device directly.

## 6.2 STREAM-BASED ACTIVE LEARNING

In these experiments, we compare our methods introduced in Section 5 with state-of-the-art methods for batch stream-based AL. All experiments in this section are conducted according to the stream batch scenario from Figure 1c. After an ablation study we select our TDLS and TEGLS method for further comparison, as state-of-the-art approaches mostly combine diversity and uncertainty as well, details can be found in Appendix C.

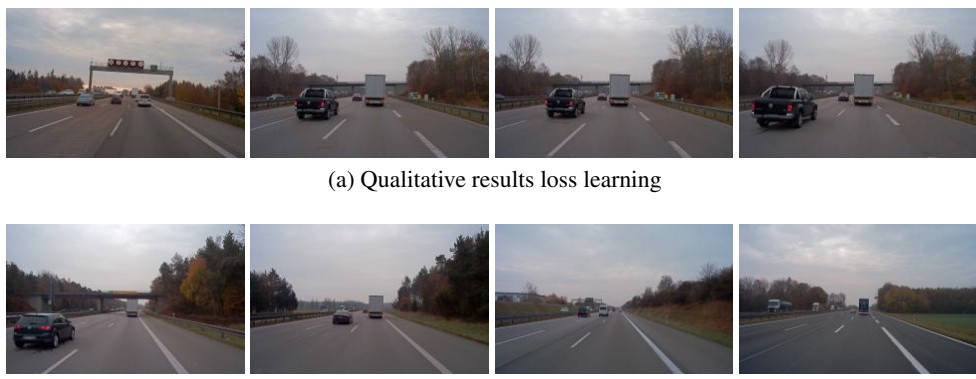

(a) Qualitative results loss learning

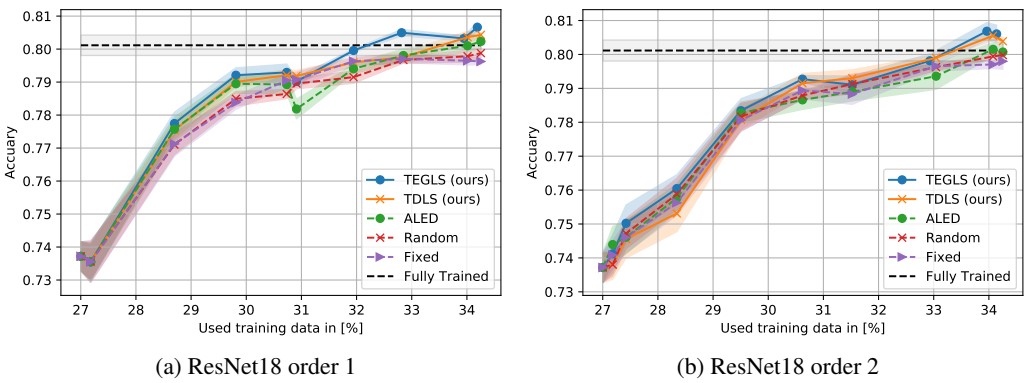

(b) Qualitative results of TPL

Figure 5: Section of selected images by TPL and loss learning in consecutive order from left to right for drive 20181108_084007.

(a) ResNet18 order 1

(b) ResNet18 order 2

Figure 6: Comparison of different stream-based methods in the stream batch scenario (Figure 1c). Colored areas indicate standard errors.

In Figure 6 we compare TDLS and TEGLS with Random, Fixed and ALED for two different orders. In Figure 6b both methods need one stream selection less to cross the fully trained network's performance at 33% and use 0.5% labels than ALED. In Figure 6a TEGLS cross the fully trained network's line as 31.8% while ALED needs about 1% more labels to achieve the fully trained networks performance at 32.8%. It can be seen that combining the temporal change of uncertainty with the most informative latent space encoding works best and outperforms random as well as state-of-the-art selection methods.

## 6.3 FROM STREAM-BASED TO POOL-BASED ACTIVE LEARNING

The main body of work in the field is focused on pool-based AL only. We want to compare the two scenarios stream batch (Figure 1c) and pool stream (Figure 1b) and close the gap. To the best of our knowledge, we are the first to compare the different scenarios explicitly. We use the stream batch and the pool stream scenario introduced in Section 3 for this experiment as otherwise, the pool-based methods could select samples from future streams. Pool-based methods can use information that is not available for stream-based methods like using data from the unlabeled pool for unsupervised training or optimization approach where the data is visited at each optimization step. Therefore pool-based methods are expected to outperform the stream-based methods. The goal of these experiments is to investigate the decrease in performance that comes along with changing from a pool-based to a stream-based setup. As pool-based scenarios require more data logistics and are computationally more expensive, a certain decrease in performance might be acceptable. Figure 7 shows the results for two different orders.

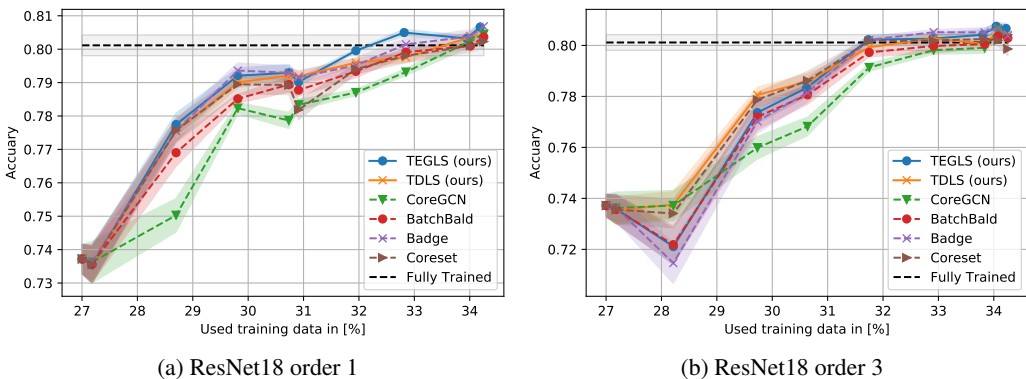

(a) ResNet18 order 1           (b) ResNet18 order 3

Figure 7: Comparison of stream-based and pool-based active learning. Pool-based methods are indicated by a dashed line. Colored areas indicate standard errors.

In Figure 7a TEGLS even reaches the fully trained network's performance one iteration before the pool-based methods and can achieve a data saving of around 1%. In Figure 7b, our method does not suffer from any performance loss due to the disadvantage in the scenarios and reaches the fully trained network's line together with other methods at 31.5% used data. The performance better than the fully trained network can be explained by the independence of drives in the different sets. So a subset can have a better distribution fit.

To the best of our knowledge we are the first showing that a stream-based method can compete with pool-based approaches in terms of performance. Our proposed method offers data savings like state-of-the-art pool-based methods, while offering the reduced data logistics of stream-based approaches. As most perception problems start from sensor (e.g. camera) streams our method can be used on the mobile device directly for many applications like robotics, autonomous driving or environmental surveillance. This saves additionally computational costs and the data prepossessing efforts.

Although our ablation study in Appendix C showed that the combination with diversity based approaches work quite well, TPL and TDLS do not need any complex matrix operations or submodular optimization techniques.

## 7 CONCLUSION AND FUTURE WORK

In our work, we investigated stream-based AL for temporally coherent data. The proposed theoretical modifications that make it possible to exploit the temporal information resulted in three classes of methods. To evaluate these scenarios, we created a classification dataset with temporally coherent data including timestamps based on the A2D2 autonomous driving dataset for this purpose, which we made publically available. In our first experiment, we showed that our modifications applied to loss learning outperform the vanilla approach by saving up to 5% more labeled data. Our second experiment proved that our methods combining temporal changes in (pseudo) uncertainty with diversity lead to 1% additional data saving in comparison to state-of-the-art methods in stream batch AL. In the last experiments conducted, we provided, to our knowledge, the first comparison between stream-based and pool-based AL using pool stream and stream batch scenarios. These experiments could prove that our stream-based methods achieve - with 0.5% more data savings - the same performance as pool-based methods for temporal data, so we bridged the gap between them. Given the additional effort, pool-based scenarios require in terms of data logistics, this is a major point for enabling large scale AL.

In future work, we want to focus on alternative uncertainty estimation methods. For one, the uncertainty estimation can be integrated more easily with a diversity measurement, which already showed good results in pool-based approaches. After we proved the benefit of exploiting temporally coherent vision data, we want to extend our approach to semantic segmentation and object detection. Additionally we plan create a dataset for steam-based in mobile robot perception.

## REPRODUCIBILITY STATEMENT

To ensure reproducibility we use the five seeds 1, 42, 64, 101 and 999 and set cudnn to "deterministic". We used the PyTorch modelzoo[2] implementation of the ResNet18 model and mention all modifications in Appendix B. An exact data preprocessing as well as training parameters can be also found in Appendix B. The exact dataset detail including all training, validation and test split as well as the split between the initially labeled and unlabeled pool are given in Appendix A.

## ETHICS STATEMENT

In our work, we deal with data selection and recording on mobile devices. Nevertheless, such recordings are necessary and already conducted, especially in autonomous driving data is collected by different institutions. To respect personal data privacy these recordings are strictly regulated by governmental authorities. We strongly encourage to respect these regulations. Nonetheless, such methods can be used to collect data unauthorized. However, we think that the benefit of our stream-based AL approach and data collection on mobile devices can create to increase the perception and reliability of these mobile devices, robots and autonomous cars outweigh this risk.

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

## A    DATASET DESCRIPTION

In our experiments, we used the object detection part of the A2D2 dataset[3]. The dataset contains 17 different drives in southern Germany. The frames are timestamped with a high frequency of up to 10 Hz so that the temporal change of the samples can be evaluated meaningfully. Due to sensor synchronization the rate is not constant, but the optical flow does not get lost. This high frequency brings the risk of selecting redundant samples in a batch. The recording drives are split into an initial labeled pool and unlabeled pool for training as well as validation and test set as shown in Table 1. In the stream-based setups, the unlabeled drives are fed as streams into the AL algorithm. The images have been resized with preserved aspect ratio to $240 \times 151$ pixels. For the training, the images have been normalized with the mean and the standard deviation of the currently selected samples. The training dataset has been shuffled.

## B    DETAILED EXPERIMENT DESCRIPTION

We used for all experiments PyTorch, for the existing reference methods we used the code published by the authors Kirsch et al. (2019)[4] and Caramalau et al. (2021)[5]. The submodular optimization approaches are implementation using Sieve-Streaming++ (Kazemi et al., 2019)[6]. The ResNet18 has been modified with a three fully connected layer classification head with inner dimensions of 256 and 128. In front of each fully connected layer, we added a dropout with a probability of 0.3. These layers remained active for the MC dropout based methods with ten forward passes. A softmax activation is attached to the last layer. For the convolutional layers the pre-trained weights provided

---

[3] https://www.a2d2.audi/a2d2/en/download.html
[4] https://github.com/BlackHC/batchbald_redux
[5] https://github.com/razvancaramalau/Sequential-GCN-for-Active-Learning
[6] https://github.com/ehsankazemi/hybrid-streaming

Table 1: This table shows the dataset split into internal labeled and unlabeled pool training set as well as validation and test set.

| Assignment | Sessions |
|---|---|
| initial labeled | 20181107_132730 20181108_091945 |
| unlabeled order 1 | 20181107_133258 20180807_145028 20180925_135056 20181107_132300 20181204_170238 20181108_084007 20180810_142822 20181008_095521 20181204_154421 |
| unlabeled order 2 | 20181204_170238 20181204_154421 20181107_132300 20181008_095521 20180925_135056 20180810_142822 20180807_145028 20181108_084007 20181107_133258 |
| unlabeled order 3 | 20181107_133258 20181108_084007 20180807_145028 20180810_142822 20180925_135056 20181008_095521 20181107_132300 20181204_154421 20181204_170238 |
| validation set | 20180925_101535 20181016_125231 20181204_135952 |
| test set | 20180925_124435 20181108_123750 20181108_103155 |

by PyTorch has been used and freezed. We trained each model with the SGD optimizer using PyTorch 1.11.0 with a learning rate of 0.0001, a momentum of 0.9 and a weight decay of 0.0005 for a maximum of 200 epochs. To ensure convergence we use early stopping on the validation accuracy with a patience of 30. The batch size is set to 128. Due to the drive concept of A2D2, the dataset contains samples very close to each and can contain only small changes in the image. So the parameters have been selected, such that the model does not overfit on the initial set. As a performance increase has been observed when training with an attached loss learning module, this module has been attached to all models. This effect was only observed for a few amounts of data. The loss learning weight in the loss function is set to 1. The learning rate is decreased by a factor of ten, after 160 epochs. The parameters for CoreGCN are taken from the authors' implementation Caramalau et al. (2021). The experiments are conducted on Nvidia V100 Graphic cards.

## C  ABLATION STUDY

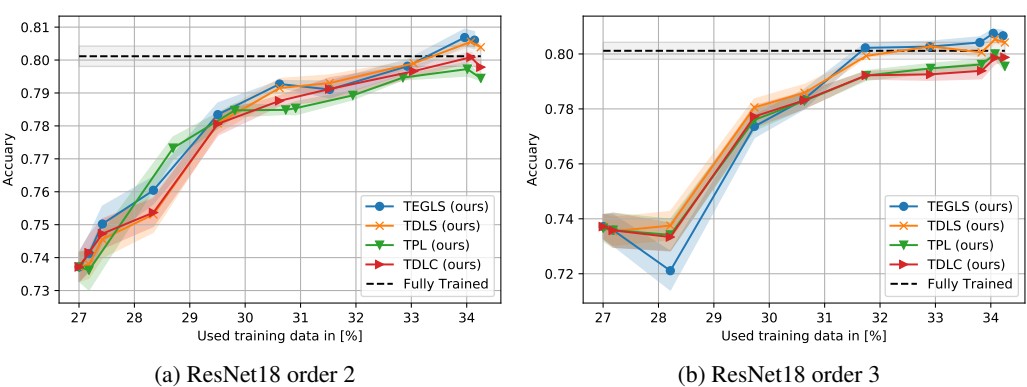

(a) ResNet18 order 2

(b) ResNet18 order 3

Figure 8: Comparison of the introduced stream-based methods in the stream batch scenario (Figure 1c). Colored areas indicate standard errors.

We compare our methods introduced in Section 5 with each other in an ablation study to investigate the effect of our proposed adaptions. In Figure 8 we show the comparison for two different stream orders. It can be seen, that the temporal change in latent space distance (TDLC) without submodular optimization, does not generate a huge improvement, only a minor one can be seen in Figure 8a. As the distance reflect already the change in latent space, the derivative of this change does not seem to add much value. However, the combination of the distances (TDLS) or embedded gradients

(TEGLS) with the temporal loss as a submodular optimization problem seems to improve our base approach.

