# OpenReview forum: "Bridging between Pool- and Stream-Based Active Learning with Temporal Data Coherence"
_ICLR.cc/2023/Conference — Submitted to ICLR 2023_

### Official Review · Reviewer_SVPd · 2022-10-17

**Confidence:** 2
**Correctness:** 3
**Technical Novelty And Significance:** 2
**Empirical Novelty And Significance:** 2
**Recommendation:** 5

**Clarity, Quality, Novelty And Reproducibility:**

Clarity:
The description of the algorithm can be improved. Section 5 describes the algorithmic details of the proposed methods, but they are not sufficient. For example, how to apply SieveStreaming++ is not clear. Since SieveStreaming++ is a method for a static set function, the objective function should be updated at an appropriate timing (minor comment: det(FF^T)+I_j should be replaced by det(FF^T+I_j)), but the authors do not address this point.

Novelty:
The novelty of this paper lies in the idea of using $\left\lVert \frac{d}{dt} \hat{\mathcal{L}} \right\rVert$ instead of the loss itself. It is a nice contribution, but not so ground-breaking.

Reproducibility:
The datasets and random seeds are addressed. However, the descriptions of the algorithms are not sufficient to reproduce the experimental results. It is helpful if the authors provide the details of the algorithms (how to estimate the loss function, when the loss estimate is updated, etc).

**Strength And Weaknesses:**

Strengths:
As mentioned in the paper, stream-based active learning is not much studied compared with pool-based active learning. However, since data are often collected in a streaming fashion (e.g. autonomous driving), the importance of stream-based active learning is increasing. This paper proposes practical ideas for stream-based active learning, and then the results might be used in future applications.

Weaknesses:
The support for the superiority of the proposed algorithms is not so strong. There is no theoretical guarantee, and why the authors choose the proposed criteria (objectives (1) and (2)) is not explained clearly. Although the authors empirically show that the proposed algorithms perform well in some object detection tasks, but only for the specific task with specific datasets. I think more experiments are needed if the authors claim that these algorithms perform well for general stream-based active learning with time stamps.

**Summary Of The Paper:**

This paper applies stream-based active learning to datasets with time stamps. The main idea is to select data points with a large loss change $\left\lVert \frac{d}{dt} \hat{\mathcal{L}} \right\rVert$, where $\hat{\mathcal{L}}$ is an estimated loss. The authors propose three methods by following this policy, and validate their empirical quality in object detection experiments.

**Summary Of The Review:**

Stream-based active learning with temporal information is an important problem and the proposed algorithms based on the loss change is practically useful, but it would be better to discuss in more detail the advantages of the proposed algorithms.

---

> ### Author Response · Authors · 2022-11-17
> **Thank you for your Feedback**
>
> Dear Reviewer SVPd,
>
> Thanks for your valuable review.
>
> **Support for the superiority:**
> We reformulated some parts in Section 5 & 6 to support the superiority of our methods and highlight that stream-based active learning can challenge pool-based active learning.
>
>
> **Experiments:**
> We try to extend our experiment until the end of the rebuttal phase with different models, however in terms of datasets we are limited to possibilities of public datasets. We have checked the most famous dataset in the domain for suitability including Kitti, Cityscapes, Nuscences, Berkely Deep Drive, but due to the focus on perception tasks itself and therefore not suitable. Our approach to even the lack of suitable data is the approach of different orders as a kind of Leave-One-Out Cross-Validation. If you have a suitable in mind, we would be more than thankful. Additionally, we have already started discussions about creating a dataset for the purpose of stream-based active learning in perception.
>
> In this field showing one task – classification - even for task-agnostic methods is common in state-of-the-art, so we decided to show one task here. e.g.
> - Kim et. al. CVPR, 2021, Task-Aware Variational Adversarial Active Learning
> - Caramalau et. al.,CVPR 2021, Sequential Graph Convolutional Network for Active Learning
> - Ash et. al. ICLR, 2020 Deep Batch Active Learning by Diverse, Uncertain Gradient Lower Bounds
>
> However, there are papers in the domain of active learning in perception showing multiple tasks, so we put this on our list, but it is challenging to achieve this until the end of the rebuttal period.
>
>
> **Clarity & Reproducibility:**
> Regarding the clarity, we updated the description to make it clearer.
> The objective function is not updated, as the objective is not updated, and conceptual drifts or other noises are neglected. As models are retrained, the representation of the samples latent space F and the predicted loss $\hat{\mathcal{L}}$ changes which makes the objective function dynamically. We put this as in Section 5 to describe the components behavior during the active learning cycles. Additionally, the set is changing based on the active learning cycle and either by increasing the pool or changing stream.
> To tackle this point and the reproducibility, we plan to publish our code, which is in our opinion more helpful than depicting an algorithm.
>
> **Novelty:**
> We appreciate your summary about the novelty “It is a nice contribution, but not so ground-breaking.”. The idea itself is not so ground-breaking, however with this idea we can show that a stream-based approach can outperform state-of-the-art pool-based active learning approach. In the field of active learning for deep learning-based perception pool-based approaches are dominating as they can use unlabeled data for semi-/self-supervised training, use an optimization globally and visit each data point as often as needed during the optimization. A majority would certainly belief with this advantage in data access pool-based methods clearly outperform stream-based approaches in terms of performance. This is certainly also reflected in the amount of works published in pool-based active learning in contrast to the amount of works in stream-based active learning in the area of deep learning-based perception.
> To the best of our knowledge, we are the first who compare stream-based with pool-base methods directly and show that there is no performance gap for stream sensor data. By showing that our stream-based approach can even outperform state-of-the-art pool-based approach using timestamps which are always present during recording and just get disposed is what makes our contribution more ground-breaking than the nice idea itself.
>
> Sincerely
> the authors of Paper4679

---

### Official Review · Reviewer_aUcz · 2022-10-20

**Confidence:** 4
**Correctness:** 1
**Technical Novelty And Significance:** 1
**Empirical Novelty And Significance:** 1
**Recommendation:** 5

**Clarity, Quality, Novelty And Reproducibility:**

**Clarity/Open Points**
- Lack in language clarity:
   - Distinction of “uncertainty” and “diversity” into “predict one value” and “performing greedy optimization” are not appropriate, why not stick with the known categorizations?
   - References missing, for example: “While early approaches like loss learning or MC dropout are generating a single value per unseen image, state of the art approaches often include unlabeled data for unsupervised training. Other approaches taking diversity into account usually perform a greedy optimization, which requires constant access to the complete unlabeled dataset.
- Correct use of language lacking, e.g., consistency of names (“pool based” vs “stream-based”, pytorch, BatchBALD vs. BatchBald, Active learning/Learning), typos. Please have the text proof-read.
- Motivation is bad: “As classification is still the most commonly benchmarked task and shown by all authors, we focus on classification in our paper.”
   - What is the task in the autonomous driving use-case that you really want to tackle with your work? Detection of highways or construction sites? Events/Incidents?
- Related work
    - If the work cited in Section 2.1 is related work, then please differentiate your method from it. If it is not (this is what I guess) then omit it (many things already have been used to motivate AL and introduced before)
    - Where is the related work section on stream-based Active Learning? Section 2.2 misses a lot of related work on stream-based AL, e.g., Fuji2016, see below.
    - If you use AL to filter a stream of data, should you not also present related work on stream processing such as novelty detection?
- Method:
    - How do you use the time? You only describe the difference in gradients and the distance in the latent space, no temporal information
    - Section 3: Why don’t you compare to “third category methods”? Is it not possible to train VAAL on stored data in a datacenter once and then apply it on a stream on device?
    - Section 4: Construction of dataset for classification: what are the problems that arise from the inconsistent sample rate? Are the frame timestamps later used in the method?
    - Section 5: You mention the method is sensitive to sudden occurrences. What do you mean?
- Claims that are not supported by experiments:
    - Noise of results in Figure 7 is high. Results shown are specific picks (here: orders) to support the claim, but do not show summary statistics. Also, results have high variance and difference between methods is small, drowned by noise.
    - Figure 4a: why are the data points not equidistant on the x-axis, if you select 5% of the samples for labeling?
    - Figure 6a,b: missing random baseline
- Conclusion: what other tasks do you want to extend the approach to?

**Related work (non-exhaustive list, as a starting point)**
- Fuji et al., 2016, Budgeted stream-based active learning via adaptive submodular maximization
- Narr et al., 2016, Stream-based Active Learning for efficient and adaptive classification of 3D objects
- DavideCacciarelli et al., 2022, Stream-based active learning with linear models
- Li et al., 2019, Incremental semi-supervised learning on streaming data
- Korycki et al., 2019, Active Learning with Abstaining Classifiers for Imbalanced Drifting Data Streams
- Luo et al., 2017, Online decision making for stream-based robotic sampling via submodular optimization

**Minor details**
- The paper talks about “1% fewer labels” or “0.5% fewer labels needed” – this was somehow confusing to me as it did sound marginal. In retrospect, I think it would be better to relate this value to the number of samples actually being annotated
- “A collection of different scenarios [is] depicted in”
- “Classification datasets like Cifar10 (Krizhevsky et al., 2009)[,] which is often used to benchmark AL[,] are not temporally ordered data streams.“
- “The scaling factor $\delta$ is chosen to 0.5” $\rightarrow$ “we set the scaling….”


**Strength And Weaknesses:**

**Strengths:**
- Addresses a relevant problem: early pre-processing of data close to sensors

**Weaknesses:**
- Language in-correctness and lack of clarity (see below)
- Motivation of the method (why is the classification of highways vs. construction sites important?)
- Claims are not well supported by experiments nor by theory


**Summary Of The Paper:**

The authors propose an Active Learning method for AL in a streaming context for an application of AL on edge (sensor) devices such as a fleet of cars or robots. Additionally, they define two contexts with different configurations of pool-based batching and streaming of data. Then, they introduce and evaluate variants of loss learning (Yoo & Kweon, 2019) on a new 4-class classification task defined on the Audi Autonomous Driving Dataset (Geyer et al., 2020) and ResNet-based architectures.

**Summary Of The Review:**

The paper addresses the field of stream-based active learning which is definitely relevant for a lot of field applications where actual systems are being deployed. However, as the paper misses a lot of related work and does not support its central claims neither theoretically nor experimentally, I do not see ICLR as an appropriate venue for this work. Moreover, writing needs to significantly improve, making this paper not mature enough at this point.

---

> ### Author Response · Authors · 2022-11-17
> **Thank you for your feedback**
>
> Dear Reviewer aUcz,
>
> thank you for your valuable feedback.
>
> First, I would like to comment on your summary. For us it is not clear how our main contribution is reflected here. Especially the properties we emphasize in Section 3 and the method we define based on the properties in Section 5.
>
> **Clarity/Open Points:**
> - Lack in language clarity:
>   - **Categorization:** We improved the description in Section 3. With this categorization we make a statement about the usability in stream-based AL, which is not reflected in the already known ones. The known ones make a statement about which kind of metric is evaluated to select samples e.g. Uncertainty or diversity.
>   - **References:** In your example you gave for missing references, the methods in the sentence are introduced in sentences above in the same paragraph by name, reference category.  We updated it such that each naming is referenced especially in consecutive following sentences.
>  - **Correct use of language lacking:** Thanks, these seem to have slipped us.
> - **Motivation:** You are quoting one sentence of the contribution description from Section 1 here. However, the motivation is built up during the whole section, mainly in the third paragraph. TLDR: Pool-based AL does not scale with the needs of open world perception; we purpose a novel concept for stream-based AL. We select a task and create a dataset suitable for comparison.
>    - **Task Motivation:** We agree that the motivation for the task of problem the newly created labels solve, we will update this. However, this is not the core of the paper and thus not the main motivation. In theory other perception tasks fulfilling the properties in the dataset could be used. Additionally, we updated the third paragraph of the introduction, which should make to motivation of creating
> You are right, we created the labels with a real problem in mind, namely operation-domain-detection. We updated this to address the importance and motivation of this task.
>
> - Related work
>   - **General Statement:** In the related work we focus on active learning for deep learning in perception. That’s why section 2.1 is much longer than section 2.2 as the main direction in this field is pool based active learning.
>   - **Section 2.1:** In section 5 we explain how we build on related work and include temporal data and derive our methods. We mention there what parts of related work we are using. We will come to that point in answer to your first question about the methods. Unfortunately, we are unsure about your intention here. Is your point that our work does not differ from the related work and section 5 should be omitted or pool-based active learning for perception using deep learning is not related work at all and the whole section should be removed?
> In this case, we argue that we are reflecting the work that has been done in active learning for perception with deep neural networks. In this field pool-based active learning is the dominant approach. It is important to reflect for understanding of our methods, especially as we aim to compare both scenarios. Could you please elaborate your point more precisely?
>   - **Section 2.2:** As mentioned above we focused on literature dealing with perception using deep learning. In this field the literature in the area of stream-based active learning is quite limited. We added literature about stream-based active learning in perception (e.g. Narr et. Al 2016) and general concepts like Fuji et. al. 2016.
>   - **Novelty detection:** We politely disagree here, as we do not touch the concepts of novelty detection. Novelty detection belongs to the family of Generalized Out-of-Distribution Detections (Yang et. Al 2021, Generalized Out-of-Distribution Detection: A Survey), so one could also add then Anomaly Detection, Out-of-Distibution Detection or other fields in which data is seen as a stream.
>
> - Method I:
>   - **Usage of time:** In Section 5 we introduce our idea of using the time derivative d/dt after a few explanation sentences. The temporal change is the key concept of the paper as it is used in every method formulation (equation 1 & 2). We updated the beginning of Section 5 to make it more obvious.

---

> > ### Author Response · Authors · 2022-11-17
> > **Thanks you for your feedback - part II**
> >
> > - Method II:
> >    - **Third category comparison:** In the experiments about the comparison between pool- and stream-based active learning we show in Figure 8a comparison with CoreGCN which is from the third category (learning-based methods using unlabeled data in the training), as mentioned in Section 3. It is more novel and outperforms VAAL. We compare methods not suitable for stream-based active learning in these experiments by using the pool batch scenario as described in Section 6.2.
> > **Pre-trained VAAL:** To your question about the usability: VAAL is not suitable for training before the deployment on a device, as the discriminator is trained by using unlabeled and labeled data as binary classification task. This requires unlabeled data initially on the data center, which cannot be assumed in a stream-based scenario. For more details and a good sketch (Figure. 1) please refer to Sinha et. al. 2019.
> >   - **Inconsistent sampling rate:** You are right here; we did not mention the downsides. We updated this in Section 3/Appendix A. TLDR: As the optical flow does not get lost and use the time derivative there are none.
> >   - **Sudden occurrences**: Sudden occurrences are for example a pedestrian just walking in the field of view and either crossing it or waiting at a traffic light etc. If the pedestrian does not disable in the next frame by making a 180-degree turn, you can roughly imagine it as an additional constant on the uncertainty like a step function. If you take the derivative, you get a Dirac delta function, which is easier to detect by maximum operation and avoids multiple selections in one plateau.
> > - Claims that are not supported by experiments:
> >   - **General statement:** It is hard for us to follow your thoughts here. We show four different plots and two figures visualizing the effect in image and latent space. The points given focus on the outline or information given in specific plots, which we hopefully answered completely below. However, we don’t see how you conclude to the emphasized point mentioned in the weaknesses.
> >   - **Noise:** With the different orders we want to balance the lake of available dataset fulfilling the properties described in section 3 in a leave one out validation fashion. We understand the concern about specific picks, however showing each time 2 out of 3 such that in total each order is shown twice, does not seem picked to us. As the streams vary in length and information content, a summary of all orders is not meaningful. We will use this critical feedback to improve the paper with the required experiments.
> > We politely disagree that the variance and difference in methods are too high and state that this can occur in this field. Furthermore, the usually used datasets in this field, namely cifar10/100, SVHN, etc. are designed for classification and provide higher diversity, than our dataset which is meant to model a sensor input stream which negatively effects the difference of the methods. .
> > We put collection paper from the field of active learning for image classification where methods overlap or have a high variance:
> > Methods overlapping in Figure 7,
> >  Yoo et. al. CVPR, 2019, Learning Loss for Active Learning
> > Methods overlapping in Figure 2,
> > Sinha et. al., ICCV, 2019, Variational Adversarial Active Learning
> > High variance overlapping's in the improvement plot, Figure 3, Kim et. al. CVPR, 2021, Task-Aware Variational Adversarial Active Learning
> > High Variance and overlaps Figure 3 & 4:
> > Caramalau et. al.,CVPR 2021, Sequential Graph Convolutional Network for Active Learning
> > High Variance and overlaps Figure 3:
> > Ash et. al. ICLR, 2020 Deep Batch Active Learning by Diverse, Uncertain Gradient Lower Bounds
> >    - **Plot axis labels:** In Section 6 we mention that nine drives of varying length remain as unseen streams and the selection size is based on the specific stream length. At each cycle (Fig 1.c) 5%/10%/20% of a stream are selected and not 5%/10%/20% of the total training set. We improved our description in Section 6.
> >   - **Random missing:** Figure 6 compares only our methods, Figure 7 compares our methods against state-of-the-art, fixed and random in the same experiment setting (stream batch). We tried not to overload the plots, that is why we split them into two in the first place.
> > - **Conclusion:** Explicitly named tasks now.

---

> > > ### Author Response · Authors · 2022-11-17
> > > **Thanks for your feedback - part III**
> > >
> > > **Related work –provided list:**
> > > - Thank you for the list of literature, it is a great overview about different applications of stream-based active learning and stream processing. Unfortunately, we lack a bit the big connections strings between some of the papers with respect to our task. For example, DavideCacciarelli et al., 2022 covers stream-based active learning based on the linear state model of a chemical plant for a regression task. Could you please explain the connections in more details?
> > > - You convinced us that our selection criteria are too strict for stream-based active learning, as deep learning-based perception is not a big topic there. But the field between chemical processes and decision making seems to be a bit wide to us. We updated the related work and hope for a fruitful discussion.
> > >
> > > **Minor details:**
> > > - We understand your point, but as the dataset we are using is quite unknown and contains a lot of redundancy we believe the percentage is more useful. Additionally, the absolute sizes are given in Section 6.
> > > - We corrected the typos
> > >
> > >
> > > Sincerely
> > > the authors of Paper4679

---

> > > > ### Comment · Reviewer_aUcz · 2022-11-22
> > > > **Re. Author Response**
> > > >
> > > > I would like to thank the authors for their response. After reading the response, the other reviews, and after reviewing the new version of the paper, I must admit that I did not totally grasp the main contribution of the paper in the first run. I am sorry for that.
> > > >
> > > > The contribution lies in the new loss variants that exploit temporal properties of the data stream. While I agree with other reviewers, that this is marginal contribution, it still makes sense and is hence makes a valuable point. Of course, it is also not 'ground-breaking' but I think it is still enough for a well-positioned paper, especially in the light of the benchmarks against pool-based methods.
> > > >
> > > > One central issue (also during the initial read) was the motivation of the application, which at least to me was not so convincing:
> > > >
> > > > *As most perception problems start from sensor (e.g. camera) streams our method can be used on the mobile device directly for many applications like robotics, autonomous driving or environmental surveillance. This saves additionally computational costs and the data prepossessing efforts.*
> > > >
> > > > How is the setup here? We have car running a perception model, identifying important samples, transfering only the important samples to the data center, which trains a new model that is then transfered back to the car? Then the car can resume driving and collecting the next frames? Or should the car train its own model? In both cases: who is labeling the data in practice and how would you integrate more cars into the pipeline from a system point of view? In the robotic use-case: how is the oracle interacting with the robot and where is the model trained? My point is: in all that scenarios a human observer needs to annotate a sample, train the model, and deploy the model, before new data can be processed. This seems impractical for most use cases which is the central argument behind many pool-based methods (and maybe the paper should argue on that or even propose an experimental design for this).
> > > > Also related to that: What if data arrives in streams but without any temporal correlation, i.e., data from different sources?
> > > >
> > > >
> > > > In its current form I still see the following weaknesses in the paper:
> > > > - The section on related work was still not well-focused. I still seem imbalanced to me as there is too much focus on pool-based AL. I have also read the reponses on that but I still do not see the point of this
> > > > - While there are improvements in AL, the results in Fig. 6 are not fully convincing, as they are only marginal
> > > > - The experiments should cover more datasets: as of now, it is unclear if the results are also generalizing to other datasets. I know this is challenging as there is not much out there. But this is still one the major issues.
> > > >
> > > > In the light of the new version of the paper. and the discussion, I will increase my score to 5, but still not fighting for acceptance of the paper due to the aforementioned weakenesses.

---

> > > > > ### Author Response · Authors · 2022-11-29
> > > > > **Re. Reviewer Response**
> > > > >
> > > > > Dear Reviewer aUcz,
> > > > >
> > > > > thank you for your response and your updated opinion.
> > > > >
> > > > > **Questions:**
> > > > >
> > > > > > How is the setup here?
> > > > >
> > > > > The setup is quite abstracted in Fig 1, so let us give an example here.
> > > > > In (Fig 1d) the classic pool-based setup dominating the field is shown. In this case a robot/car/device needs to record as much data as possible which is then transferred to the data center. There the selection path happens. Selected samples are then sent to a labeling company. In the case of research experiments the labels are present, and the process is just simulated. After the samples have been labeled the model is retrained. In our proposed case (Fig 1b) the selection is shifted from the data center to the robot/car/device which reduces the data logistics effort in transferring this data to a data center and makes it scalable with the number of cars/robots. In Fig 1c, the selection is performed on the data center, but before all data is collected. A selection cycle is done after each drive.
> > > > >
> > > > > In any scenario the models are trained on the data center which is mostly done for deep neural networks thus the classic stream-based scenario (Fig 1a) is not present in literature for deep neural networks. How the model is transferred back to a car/robot/device can be arbitrary, in our case we assume a data upload after each drive. We assume that before the next drive the model will be updated, which might not fit all real-world conditions (training times etc... ). However, this affects all active learning scenarios.
> > > > > The integration of multiple agents and arising problems is neglected in our work for now.
> > > > >
> > > > > >My point is: in all that scenarios a human observer needs to annotate a sample
> > > > >
> > > > > About your highlighted point, in contrast to the pool-based approach we just exchange where the data is selected, and the amount of data needs to be transferred.  Also, in a pool-based approach, the data needs to be transferred from the robot to the data center, the data needs to be annotated by some service provided, the models need to be trained on a data center and deployed on the robot again. This comes with the properties of active learning for perception. In our work we tackle the amount of data which needs to be transferred and distribute the selection, the rest remains unchanged. Your point is correct for classic pool-based active learning where all data is already present at the data center, labeling cycles are fast and the amount of data is low. On the one hand, we can save the downloads of the model after each cycle. On the other side we still need to transfer first all drives to the data center, wait until all drives are finished and should know that our dataset is then complete. In real-world perception problems, as mentioned in Section 1, this is hardly the case. For example, an autonomous car produces several terabytes a day (exact number differs here between the setups). Additionally, the transfer of the data from the robot/car/device and the download of the updated model is less effort than the process of labeling or getting labels from a label provider. As, this is the critical path, we do not see any disadvantage here, as our proposed scenario does not change the critical path.
> > > > >
> > > > > > What if data arrives in streams but without any temporal correlation
> > > > >
> > > > > In the robotic domain, including autonomous driving, sensor synchronization is a known task, as we limit our work for sensors used in this field, temporal correlation and synchronization can be ensured.
> > > > >
> > > > >
> > > > > **Weaknesses:**
> > > > > - Related work: Here we are wondering, we explained our focus which results in an imbalance between stream-based and pool-based active learning. The point of focused you did not mention before. When we got you correctly, you are suggesting balancing the related work more in the direction of stream-based active learning, which will result is less focus as the number of works in the perception domain dealing with stream-based active learning is low. This seems to be contradicting with a clear focus in the direction of our topic/domain.
> > > > >
> > > > > - Experiments and dataset: We are working on the experiments, but could not finished them until the end of the rebuttal period. Additionally, we would be more than thankful if you know by chance a dataset. Currently, we think that adding more tasks will enable us to use more datasets.
> > > > >
> > > > > Sincerely
> > > > > the authors of Paper4679

---

### Official Review · Reviewer_63pA · 2022-10-30

**Confidence:** 3
**Correctness:** 3
**Technical Novelty And Significance:** 2
**Empirical Novelty And Significance:** 2
**Recommendation:** 5

**Clarity, Quality, Novelty And Reproducibility:**

- The paper gives a view on how to transform pool-based methods into stream-based methods. It shows that using temporal changes indeed gives a more diverse sampling strategy than standard pool-based methods.

- The explanation of the implementation of temporal predicted loss could be improved by supplying an algorithm description.  Very clear explanation of the second method, why it is hypothesized to work well, and how it should be implemented. The TDLS method would be easier to reproduce if a step-by-step algorithm description would be supplied.


**Strength And Weaknesses:**

+ The paper is written clearly and easy to follow.
+ The paper gives a view on how to transform pool-based methods into stream-based methods.
+ The custom dataset can be easily reproduced by the dataset description or downloaded as it is publicly available, and all hyperparameters needed to replicate the results are given.

- related work section can be improved a lot as it does not cover the current state of the art. Good literature overview of existing pool-based query methods and their (dis)advantages.
An overview of the properties of stream-based data, as opposed to pool-based data, is missing. For example, the non-stationary nature and possible concept drift of stream-based data are very important properties that can be solved by (stream-based) Active Learning.

- Paragraph 2.2 only contains one Stream based active learning technique. There are many more that could be discussed to give a better view of previous research. For example, it would be good to add some info about “RAL - Improving stream-based AL by RL”. There is much more research on stream-based active learning than this paper suggests.

- The obtained results can be discussed further to show the real value of the proposed approach. In the current form, it is very hard to see why the proposed approach required 1% less data than other ALs (sec. 6.2, and 6.3).

**Summary Of The Paper:**

This paper shows several methods on how to use temporal information in stream-based active learning. It shows why classical pool-based AL methods cannot be used in this domain. It also shows that some methods can even outperform pool-based methods in terms of data savings. This paper closes the gap between pool and stream-based active learning. The authors performed several experiments on the public Audi Autonomous Driving Dataset and showed marginal improvements over state-of-the-art approaches by using 1% fewer labels.

**Summary Of The Review:**

Overall this is a concise paper but some parts could be revised to further enhance reproducibility.
Algorithm descriptions would be very helpful for the reader. While some related research on stream-based AL is given it does not give a correct representation of the actual research that has been done in this domain. The obtained results should be discussed in more detail to show the usefulness of the proposed approach. In its current form, it is hard to judge its value.

---

> ### Author Response · Authors · 2022-11-17
> **Thank you for your feedback**
>
> Dear Reviewer,
> thank you for your valuable feedback.
>
> We appreciate your positive and critical comments.
>
> **Related work:**
> In the related work we focused on active learning for deep learning perception, which resulted in few literatures in Section 2.2. We updated the related work to also include stream-based active learning from non-deep learning and non-perception domains to give the reader a better overview.
>
> **Discussion:**
> Additionally, we made our discussion in 6.2 and 6.3 clearer and extended it a bit, to leverage the advantages of the proposed method.
>
>
> **Clarity:**
> We updated the description of the temporal loss and extended the general introduction in Section 5. For the reproducibility of the TDLS we plan to publish the code. We think this could be more helpful than a step-by-step algorithm description.
> We hope to improve the quality of the paper based on your suggestions.
>
> Sincerely
> the authors of Paper4679

---

> > ### Comment · Reviewer_63pA · 2022-11-19
> > **feedback**
> >
> > Thank you for clarifying the concerns! It is an interesting read! I believe almost all the necessary ingredients are there, and revising the paper more carefully to include all of the concerns mentioned by all reviewers will improve the paper significantly.

---

### Author Response · Authors · 2022-11-18
**Updates from your feedback**

Dear Reviewers,

thank you for your valuable feedback.
We updated the paper and incorporated your feedback.

**Summary of the updates:**
- We updated and strengthened the motivation in Section 1.
- We extended the Section 2.2 related work of stream-based active learning.
- The category description in Section 3 was updated.
- In Section 4 the task description was improved and some details were moved to the Appendix.
- We improved and extended the description of TPL in Section 5.
- The description of the stream ingest to the active learning cycle in Section 6 was improved.
- Result discussions in Sections 6.2 and 6.3 were clarified.
- We precised our future work in Section 7.

**Minor updates:**
- We corrected some typos and did minor updates in some sentences.
- We restructured the defined stream orders for a better reading flow in the new structure of the paper.
- The experiment and parameter description in Appendix B was updated.
- The x-axis description in the plots was improved.
- We updated plots for BatchBald, as there was a minor error found and fixed in the implementation.

Unfortunately, we could not fulfill the requests for experiments yet.


Sincerely
the authors of Paper4679

---

### Decision · Program_Chairs · 2023-01-20

**Decision:**

Reject

**Justification For Why Not Higher Score:**

All three reviewers have concerns with either the experiment part a bit weak or lacking a theory. The work is also a bit weak in elucidating the motivation.

**Justification For Why Not Lower Score:**

N/A

**Metareview: Summary, Strengths And Weaknesses:**

The paper studies active learning in a stream setup, which assumes to be more useful than the classic pool setup for applications such as autonomous driving and robotics. All three reviewers are slightly negative about the work. This submission has clear values, but overall is a bit immature to be accepted now. More experiments or theories can be added to increase its strength, and the exposition can be made more clear (especially the motivation).